# Fractionation of Potentially Toxic Elements (PTEs) in Urban Soils from Salzburg, Thessaloniki and Belgrade: An Insight into Source Identification and Human Health Risk Assessment

**DOI:** 10.3390/ijerph18116014

**Published:** 2021-06-03

**Authors:** Pavle Pavlović, Thomas Sawidis, Jürgen Breuste, Olga Kostić, Dragan Čakmak, Dragana Đorđević, Dragana Pavlović, Marija Pavlović, Veljko Perović, Miroslava Mitrović

**Affiliations:** 1Department of Ecology, Institute for Biological Research “Siniša Stanković”—National Institute of Serbia, University of Belgrade, Bulevar Despota Stefana 142, 11060 Belgrade, Serbia; olgak@ibiss.bg.ac.rs (O.K.); dragan.cakmak@ibiss.bg.ac.rs (D.Č.); dragana.pavlovic@ibiss.bg.ac.rs (D.P.); marija.pavlovic@ibiss.bg.ac.rs (M.P.); veljko.perovic@ibiss.bg.ac.rs (V.P.); mmit@ibiss.bg.ac.rs (M.M.); 2Department of Botany, Aristotle University of Thessaloniki, 54124 Thessaloniki, Greece; sawidis@bio.auth.gr; 3Department of Geography and Geology, University of Salzburg, 5010 Salzburg, Austria; juergen.breuste@sbg.ac.at; 4Institute of Chemistry, Technology and Metallurgy—National Institute of the Republic of Serbia, University of Belgrade, Njegoševa 12, 11000 Belgrade, Serbia; dragadj@chem.bg.ac.rs

**Keywords:** urban soils, potentially toxic elements (PTEs), sources of PTEs, sequential extraction, mobility, health risk assessment

## Abstract

Concentrations of potentially toxic elements (PTEs) (Al, As, Cd, Cr, Cu, Ni, Pb, and Zn) were measured in topsoil samples collected from parks in the cities of Salzburg (Austria), Thessaloniki (Greece), and Belgrade (Serbia) in order to assess the distribution of PTEs in the urban environment, discriminate natural (lithogenic) and anthropogenic contributions, identify possible sources of pollution, and compare levels of pollution between the cities. An assessment of the health risks caused by exposure to PTEs through different pathways was also conducted. The study revealed that, with the exception of Pb in Salzburg, levels of PTEs in the soils in polluted urban parks were higher than in unpolluted ones, but still lower than those recorded in other European soils. Results of sequential analyses showed that Al, Cr, and Ni were found in residual phases, proving their predominantly lithogenic origin and their low mobility. In contrast, the influence of anthropogenic factors on Cu, Pb, and Zn was evident. Site-dependent variations showed that the highest concentrations of As, Cu, Pb, and Zn of anthropogenic origin were recorded in Salzburg, while the highest levels of Al, Cr, and Ni of lithogenic origin were recorded in Belgrade and Thessaloniki, which reflects the specificity of the geological substrates. Results obtained for the health risk assessment showed that no human health risk was found for either children or adults.

## 1. Introduction

The loss and degradation of urban habitats can be linked to ongoing worldwide urbanisation [1]. Urban soils differ greatly from natural ones, given the fact that they are located in areas of intense human activity, which results in pollution, physical disturbance, and surface transformation. They are also intensively managed, and whole soil profiles are often man-made, sometimes with fertile agricultural soils from rural areas [2] or with soils of unknown origin [3]., Therefore, urban soils are considered technogenic soils in those cases where the human impact on their structure is greater than that of natural processes [4].

Critical contaminants of urban environments include persistent toxic substances, such as potentially toxic elements (PTEs), which have drawn wide attention due to their persistence in the environment, their tendency to bioaccumulate in the food chain, and their toxicity to humans and other organisms [5,6]. PTEs in urban soils often originate from multiple sources, which include natural (lithogenic) and various anthropogenic sources [7,8]. However, investigations into PTE contamination of urban soils have indicated that such contamination is mainly from anthropogenic sources, including local industrial emissions (such as power plants, coal combustion, metallurgical industries, chemical industries, etc.), vehicle emissions (e.g., exhaust particles), household emissions, pavement surfaces, building work, and atmospheric deposition [8,9,10,11,12,13,14,15,16,17]. These sources lead to a continuous increase in PTEs in urban soils, resulting in their persistence in soils for many years, even after the pollution sources have disappeared. In PTE studies, the elements As, Cu, Cr, Pb, etc., are of great concern because of their high toxicity and the potential risk they pose to both urban ecosystems and human health [5,18,19,20]. This is why investigating the distribution of PTEs in soils and differentiating their sources could offer an ideal means of monitoring and assessing soil pollution itself and overall environmental quality as reflected in soils [21,22,23].

From an environmental standpoint, it is especially important to know which natural conditions can lead to the release of various PTEs, thus posing a risk to the surrounding soil [24]. Their total content reflects the geological substrate and weathering and, in urban and industrial areas, the anthropogenic input of metals from industrial processes [25,26,27]. However, the effects of PTE exposure from soil on ecology, the environment, and health depend on the mobility and availability of the elements [28]. To address this, selective chemical extraction methods can be used to estimate the relative mobility of elements in soils, including the extent to which elements may be bound to particular soil fractions [24,29]. Factors that control the oxidation state and, thus, the mobility and toxicity of many elements, are pH, sorbent nature, the presence and concentration of organic and inorganic ligands, root exudates and nutrients, and redox reactions [28]. The mobile fractions of metals are the most important from an ecological point of view, since they are easily adsorbed and can be rapidly released with a slight increase in ionic strength or a reduction in pH.

In this study, PTE content in soils from polluted and unpolluted sites was measured in three European cities: Salzburg, with low industrial activity; Belgrade, with heavy industry and traffic; and Thessaloniki, with industry and dense traffic. The mobility and bioavailability of Al, As, Cd, Cr, Cu, Ni, Pb, and Zn was assessed via the fractionation of these elements using the modified BCR sequential extraction protocol, a preliminary assessment of the environmental and human health risk associated with PTE contamination was carried out, and the levels of pollution between cities were compared.

## 2. Materials and Methods

### 2.1. Study Sites

Sampling was conducted at both polluted (city centres) and unpolluted sites (away from the city centre, without direct sources of pollution) in three European cities: Salzburg (Austria), Thessaloniki (Greece), and Belgrade (Serbia), as shown in Figure 1.

Salzburg (47°48′ N 13°02′ E; 436 m above sea level (ASL) covers an area of 65.67 km^2^, and is located next to the German border. The number of inhabitants in the city is 152,367. Land cover is diverse in this region, with industrial, commercial, residential, and agricultural areas, forests, intensive and extensive grassland, parks, urban green areas, and domestic gardens. Salzburg has always been an administrative town with virtually no industry apart from the wood processing and food industries. There are also different types of natural protected areas (natural parks, nature protection areas, landscape conservation areas, and Natura 2000 areas). The climate of the Salzburg area is temperate, with a mean annual temperature of 8.9 °C, an annual minimum temperature of 1.3 °C in January, and a maximum of 18.3 °C in July. Total annual precipitation is 1169 mm, with the highest precipitation (153 mm) in August and the lowest (59 mm) in February. Kurgarten (2 ha), a park characterised by dense traffic, right in the city centre, was selected as the polluted site, while Hellbrunn Park (2.5 ha) was selected as the control site. This latter park is situated about 3 km outside the city, close to Mount Untersberg.

Thessaloniki (40°62′ E 22°95′ N; 0–250 m ASL) is one of the most densely populated cities in Greece, with approximately 1.1 million inhabitants (665 inhabitants per km^2^), and covers an area of 1456.68 km^2^. Activities such as oil refining, petrochemical, acid, and fertilizer production, cement production, ceramics, non-ferrous metal smelting, iron and steel manufacturing, scrap metal incineration, and fossil fuel burning are major industrial particle-emitting sources. The climate in the Thessaloniki area is temperate Mediterranean, with hot and humid summers. Mean monthly temperatures range from 5.5 °C (in January) to 28 °C (in August). Annual precipitation is 351.6 mm, with the highest precipitation (70 mm) in October and the lowest (only 1.2 mm) in July. Prevailing meteorology is characterised by weak winds (0.5–3 m s^−1^) of south-western and north-western origin. Aristotelous Square (2 ha), Thessaloniki’s main park, located in the western part of the city and crossing three main streets, was selected as the polluted site, while Lagadas (2 ha), a park situated in a primarily rural region surrounded by a sink-like mountain where two lakes lie, 15 km from the city centre on the northeast side, was selected as the control site.

Belgrade (44°49′ N 20°27′ E; 117 m ASL), the capital of Serbia and its largest city, with an area of 359.92 km^2^, is located at the confluence of the Sava and Danube rivers. With a population of 1.7 million inhabitants, it is one of the largest cities in Southeast Europe. Within a distance of 50 km to the west, there are three coal-fired power plants with the highest emissions of SO_2_, to the east there is a nitric fertilizer factory and an oil refinery, and to the south there is an iron smelting plant. However, pollution conditions have changed over the past decade, as numerous industrial facilities that were once located in the central parts of the city have ceased operations, reducing pollution considerably. Nevertheless, two issues remain with regard to pollution levels: firstly, heavy traffic, and secondly, district heating plants that operate during the winter months, resulting in elevated concentrations of SO_2_, NO_2_, Ni, Cr, and As particulate matter. Belgrade’s climate is temperate continental, with a mean annual temperature of 11.7 °C (a maximum temperature of 22.1 °C in July and a minimum temperature of 1.2 °C in January). Total annual precipitation is 669.5 mm (with the highest in November (131.5 mm) and the lowest in April (3.8 mm)). Prevailing meteorology in autumn and winter is characterised by the ”Košava” wind (25–43 m/s) of south-eastern origin, whereas winds of north-western origin prevail in summer. Hall Pioneer Park (3.3 ha), located in the vicinity of Belgrade’s former industrial zone and exposed to heavy traffic, was selected as the polluted site, while Košutnjak forest (with a sampling site covering 2 ha), located within a zone of natural *Quercus frainetto* and *Quercus cerris* forest, with no direct sources of pollution, 10 km from the city centre, was selected as the control site.

### 2.2. Sampling and Analytical Design

Soils from Salzburg (Austria), Thessaloniki (Greece), and Belgrade (Serbia) were studied. A harmonised sampling regime was applied by all those participating in the research. Soil samples for PTE analysis were collected randomly from 5 individual sampling points at each site (park), at a depth of 0–20 cm. A composite bulk sample was obtained by mixing five subsamples at each sampling point within the sampling site, making five bulk samples per sampling site. A total number of 30 composite soil samples were collected for this work: 15 from polluted areas in the urban parks, and 15 from unpolluted areas. Soils were air dried and passed through a <2 mm stainless steel sieve before analysis.

Soil texture was determined using the Atterberg method of sedimentation, with a combined pipette technique, in 0.4 M tetrasodium diphosphate (Na_4_P_2_O_7_), while the texture classes were determined using the International Soil Texture Triangle [30] and presented as percentages of sand, silt, and clay (%). Soil pH was measured in suspension (1:2.5 soil–water ratio) using a WTW (Germany) inoLab 7110 pH meter with glass electrode. Water-soluble salts (EC (dSm^−1^)) were measured using a Knick (Germany) Portamess 911 Conductometer. The soil organic matter content (%) was determined by means of a titration method, using (NH_4_)_2_Fe(SO_4_)_2_ × 6H_2_O, after digestion of samples with a dichromate–sulphuric acid solution, based on Simakov’s modification of the Turin method [31].

Sequential extraction was performed using the optimised four-step extraction procedure recommended by BCR [32,33]. Element concentrations in different geochemical phases (acid-soluble/exchangeable (F1), reducible (Fe- and Mn-oxide-bound) (F2), oxidisable (organic-matter-bound) (F3), and residual (lattice-bound) (F4) fractions) were analysed using inductively coupled plasma optical emission spectrometry (ICP–OES, Spectro Genesis, Spectro-Analytical Instruments GmbH, Kleve, Germany). The detection limits for the analysed elements in soil samples were as follows (mg kg^−1^): Al—0.005, As—0.06, B—0.05, Cd—0.001, Cr—0.01, Cu—0.01, Mn—0.01, Ni—0.009, Pb—0.02, and Zn—0.005. The accuracy of the obtained results was checked by analysing the certified sediment reference material (BCR 701) obtained from the Community Bureau of Reference (BCR, Geel, Belgium). Recovery rates for the heavy metals in the standard reference materials ranged from 84.1–107.2%. Total concentrations of As, B, Cr, Cu, Mn, Ni, and Zn in samples were estimated by adding up the concentrations in all four BCR fractions.

### 2.3. Health Risk Assessment

A health risk assessment recommended by the US Environmental Protection Agency [34] was performed in order to establish both non-carcinogenic and carcinogenic effects on the human population from exposure to PTEs in urban soil. There are three pathways of exposure to PTEs that are often used to evaluate the risk to the population: directly through oral ingestion (CDIing), inhalation through the respiratory apparatus (CDIinh), and dermal absorption through the skin (CDIder) [35,36,37]. Both children and adults were taken into consideration when calculating the hazard quotient (HQ), hazard index (HI), carcinogenic risk (CR), and total carcinogenic risk (TRC). The cancer risk was calculated in accordance with the probability of developing cancer over an individual’s lifetime [37]. The hazard quotient represents the ratio of chronic daily intake (CDI, mg kg^−1^ d^−1^) to the reference dose (RfD, mg kg^−1^ d^−1^) [36], while HI designates the sum of the HQs for each PTE and their exposure routes [36,37,38].

The risk assessment was conducted using Equations (1)–(4), as proposed by Baltas et al. [35], Hu et al. [36], Čakmak et al. [37], and Wang et al. [38], respectively:(1)$${\mathrm{CDI}}_{\mathrm{Ing}}=\frac{C\times \mathrm{IngR}\times \mathrm{EF}\times \mathrm{ED}}{\mathrm{BW}\times {\mathrm{AT}}_{\mathrm{nc}}}\times 10^{-6}\;$$
(2)$${\mathrm{CDI}}_{\mathrm{Inh}}=\frac{C\times \mathrm{InhR}\times \mathrm{EF}\times \mathrm{ED}}{\mathrm{PEF}\times \mathrm{BW}\times {\mathrm{AT}}_{\mathrm{nc}}}$$
(3)$${\mathrm{CDI}}_{\mathrm{der}}=\frac{C\times \mathrm{SA}\times \mathrm{AF}\times \mathrm{ABS}\times \mathrm{EF}\times \mathrm{ED}}{\mathrm{BW}\times {\mathrm{AT}}_{\mathrm{nc}}}\times 10^{-6}$$
(4)$${\mathrm{CDI}}_{\mathrm{Total}}={\mathrm{CDI}}_{\mathrm{Ing}}+{\mathrm{CDI}}_{\mathrm{Inh}}+{\mathrm{CDI}}_{\mathrm{der}}$$
where C represents the concentration of the PTE in soil (mg kg^−1^); IngR indicates the soil ingestion rate (200 mg day^−1^ for children and 100 mg day^−1^ for adults); EF represents the exposure frequency (75 mg year^−1^ for recreations); ED is the exposure duration (6 years for children and 30 years for residents); BW represents body weight (42.4 kg for children and 70.8 kg for adults); AT_nc_ is the average time for non-carcinogenic effects (ED × 365); InhR indicates the soil inhalation rate (7.63 m^3^ d^−1^ for children and 20 m^3^ d^−1^ for adults); PEF is the soil-to-air particulate emission factor (1.36 × 10^9^ m^3^ kg^−1^); SA is the skin surface area available for exposure (2800 cm^2^ event^−1^ for children and 5700 cm^2^ event^−1^ for adults); AF is the soil-to-skin adherence factor (0.2 mg cm^−2^ for children and 0.07 mg cm^−2^ for adults); and ABS represents the dermal absorption factor (0.03 for As and 0.001 for other PTEs, unitless).

#### 2.3.1. Non-Carcinogenic Risk Assessment

The hazard quotient (HQ) for each element, as well as the total hazard index (HI), was determined according to equations recommended by the US Environmental Protection Agency [34], Wang et al. [38], Baltas et al. [35], and Hu et al. [36]:(5)$$\mathrm{HQ}=\frac{{\mathrm{CDI}}_{\mathrm{Total}}}{\mathrm{RfD}}$$
(6)$${\mathrm{HI}}_{\mathrm{Total}}=\sum \mathrm{HQ}$$

According to a US EPA report [39], HI values below 1 indicate no negative impact on an individual’s health, while values over 1 indicate a possible negative health effect [35,36].

#### 2.3.2. Carcinogenic Risk Assessment

In order to evaluate the cancer risk (CR) of each PTE and the total cancer risk (TCR), the CDI values obtained were multiplied by the cancer slope factors (SF kg d^−1^ mg^−1^) [34,35,36,38]:(7)CR=CDI×SF
(8)TCR=∑CR

A TCR value in the range of 1 × 10^−6^–1 × 10^−4^ implies an acceptable total risk [35,36,39]; risk values below 1 × 10^−6^ indicate no significant health effects, while values that exceed 1 × 10^−4^ indicate unacceptable carcinogenic health risks for humans [35,36]. Explanations and reference values for all of the indices used in the above equations are described in detail in research by Baltas et al. [35], Hu et al. [36], and Wang et al. [38].

### 2.4. Statistical Analysis

One-way analysis of variance (ANOVA) was performed in order to test the differences in PTE levels in the examined soils (subsequent tests of normality using the Shapiro–Wilk W test and Levene’s test of homogeneity of variances showed non-significant values for all the reported ANOVA breakdowns). A multivariate statistical factor analysis technique—cluster analysis (CA)—was performed using SPSS statistical software in order to identify groups or clusters of similar sites on the basis of similarities within a class and dissimilarities between different classes [33]. Furthermore, principal components analysis (PCA) was used to assess relationships between PTE sources in soil samples from different sites.

## 3. Results and Discussion

### 3.1. Urban Soil Properties

Urban soils are responsible for a number of ecosystem services, including the recycling of organic matter and mineral nutrients, and plant growth, and contribute to those services provided by urban green spaces, from the mitigation of the urban heat island effect, to recreational services [1]. However, they are often degraded by urbanisation and polluted by urban traffic and industry. Problems associated with such soils include a high stone content and considerable differences in bulk density, particle shape, and soil structure performance [40].

In this study, the main physicochemical parameters determined for urban topsoils from Salzburg, Thessaloniki, and Belgrade—including pH, soil texture, and organic matter content—exhibited a very broad interval of variation between the examined sites (Table 1). The data show that pH values across the entire study area ranged from 6.78 (Hellbrunn, Salzburg) to 8.47 (Hall Pioneer, Belgrade), which classifies these soils as neutral and alkaline [30], confirming earlier findings that the pH of urban soils is mostly in the alkaline range [41,42].

Soil pH plays an important role in controlling the mobility of metals from soil to plants. It is widely accepted that element mobility and bioavailability increases at low pH [43,44], whereas alkaline soil conditions immobilise potentially toxic, labile forms, resulting in their low bioavailability [45,46,47]. This is supported by the results obtained in this study, particularly at the sampling sites in Salzburg (pH < 7), where levels of Al, Cr, Cu, and especially Pb extracted in the first three fractions (acid-soluble/exchangeable, reducible (Fe- and Mn-oxide-bound), and oxidisable (organic-matter-bound)) of sequential extraction were higher in soil samples (at both the control and polluted sites) than in the samples from Thessaloniki and Belgrade (Table 2, Figure 2). Alkaline conditions are induced by parent material, time, relief, topography, climate, and organisms, as well as rich-liming materials in the soil itself, which can change element speciation and activities [42]. Specifically, alkaline sites with pH values of more than 7.2 occupy less than 4% of the Tyrol, Salzburg, and Upper Austria regions, whereas strongly acidic sites with pH values below 4.5 are found in the Salzburg area (38%) [48].

In addition, other soil properties (e.g., clay and soil organic matter) and adsorption and complexation mechanisms may influence element bioavailability, especially in soils contaminated by anthropogenic activities [43,49]. In the fine-grained soil fraction, toxic element adsorption tends to be stronger than in the coarse-grained soil fraction, since it comprises soil particles with larger surface areas [50]. In our study, the content of organic carbon (OC) in the examined soil samples ranged from 0.88% to 4.16%, with the lowest values measured at both sites in Thessaloniki and the highest at both sites in Salzburg (>4%). The examined soil samples were composed predominantly of sand, with the highest content measured in samples from Thessaloniki (62–63%), followed by samples from Salzburg (44–59%), while the lowest content was measured in samples from Košutnjak forest in Belgrade (20%). Amounts of clay also varied among the soil samples: the lowest total clay fraction content (7.9–6.6%) was measured in the Hellbrunn and Kurgarten parks in Salzburg, while the highest (26–30%) was in Belgrade. The silt fraction ranged from 20% at Hall Pioneer Park up to 50% at the control site in Belgrade.

### 3.2. The Content of Potentially Toxic Elements in Selected Urban Soils

The extent of urban soil contamination was studied based on the presence of potentially toxic elements (Al, As, Cd, Cr, Cu, Ni, Pb, and Zn) in the examined soils. Table 2 shows the mean values and standard deviations of the selected elements and the differences between the sampling sites. As concentrations of Cd were below the detection level in all of the analysed samples, they are not discussed any further. In this study, in order to estimate the environmental status of the investigated soils, two guidelines were used: the mean values of the background concentrations of toxic metals in two soil types common worldwide, from the work of Kabata-Pendias and Pendias [43]; and the background values in European soils as proposed by the European Commission [51]. These guidelines were used due to the fact that PTE concentrations in soil were presented in relation to the textural composition of the soil and its pH, which are similar to the soils investigated in this study. On the basis of the results obtained, it can be concluded that the PTE content at all of the polluted and unpolluted sites is at the level of the background values proposed for European soils and, in this sense, they can be considered to be unpolluted (Table 2) [51]. In general, a higher content of the analysed elements was measured at the polluted sites, with the exception of Al and As at Lagadas, and Al, Cr, Cu, Ni, and Pb at Hellbrunn. Significant differences were found between the unpolluted and polluted sites (*p* < 0.001), with the exception of Cr (*p* < 0.05), Cu (ns), and Zn (*p* < 0.01), Table 2. Among the unpolluted sites, the highest content of As, Cu, Pb, and Zn was measured in samples from Hellbrunn Park, the highest levels of Al and Ni were measured in Košutnjak, and the highest Cr content was measured in Lagadas. Among the polluted sites, the highest content of As, Pb, and Zn was measured in samples from Kurgarten, while the highest levels of Al and Cr were measured in Hall Pioneer Park. Higher levels of Cu and Ni were measured at Aristotelous Square.

**Table 2 ijerph-18-06014-t002:** Content of Al, As, Cr, Cu, Ni, Pb, and Zn in the examined urban soils (mg kg^−1^).

Unpolluted Sites	M (SD)	*p* Value	Polluted Sites	M (SD)	*p* Value	^a^ Global Range	^b^ EU Soils
**Aluminium**							
Hellbrunn	10,359.87 (1081.60)	***	Kurgarten	8175.52 (1120.97)	***	**10,000–40,000**	**-**
Lagadas	12,193.10 (705.73)		Aristotelous	10,927.58 (943.19)			
Košutnjak	18,199.40 (1687.75)		Hall Pioneer	19,432.80 (1879.06)			
**Arsenic**							
Hellbrunn	4.10 (0.43)	***	Kurgarten	4.11 (0.56)	***	**4.4–8.4**	**-**
Lagadas	2.83 (0.16)		Aristotelous	2.61 (0.22)			
Košutnjak	2.60 (0.24)		Hall Pioneer	3.09 (0.30)			
**Chromium**							
Hellbrunn	21.63 (2.26)	*	Kurgarten	20.91 (2.86)	***	**47–51**	**50–100**
Lagadas	26.06 (1.51)		Aristotelous	32.16 (2.77)			
Košutnjak	25.55 (2.37)		Hall Pioneer	32.30 (3.12)			
**Copper**							
Hellbrunn	42.54 (4.44)	***	Kurgarten	38.33 (5.25)	ns	**13–23**	**50–100**
Lagadas	34.80 (2.01)		Aristotelous	39.66 (3.42)			
Košutnjak	26.46 (2.45)		Hall Pioneer	34.91 (3.37)			
**Nickel**							
Hellbrunn	27.55 (2.87)	***	Kurgarten	23.05 (3.16)	***	**13–26**	**30–70**
Lagadas	24.30 (1.40)		Aristotelous	40.65 (3.51)			
Košutnjak	33.02 (3.06)		Hall Pioneer	37.46 (3.62)			
**Lead**							
Hellbrunn	**109.27** (1.41)	***	Kurgarten	87.70 (12.02)	***	**22–28**	**50–100**
Lagadas	36.10 (2.09)		Aristotelous	42.74 (3.69)			
Košutnjak	45.34 (4.20)		Hall Pioneer	53.31 (5.15)			
**Zinc**							
Hellburnn	54.08 (5.64)	**	Kurgarten	66.24 (9.08)	**	**45–60**	**150–200**
Lagadas	46.22 (2.67)		Aristotelous	48.32 (4.17)			
Košutnjak	41.43 (3.84)		Hall Pioneer	61.65 (5.96)			

ANOVA, mean (SD), *n* = 5, levels of significance: *** *p* < 0.001, ** *p* < 0.01, * *p* < 0.05, ns—not significant. ^a^ Levels above the global range are underlined [43]. ^b^ levels above the EU reference values are denoted in bold [51].

As, Al, and Cr concentrations generally exhibited lower levels in comparison to the global values for soils, while Cu, Ni, Pb, and Zn were enriched to varying extents: Ni at the unpolluted sites of Hellbrunn and Košutnjak and at the polluted sites of Aristotelous Square and Hall Pioneer Park, Cu and Pb at all of the examined sites, and Zn at the polluted sites of Kurgarten and Hall Pioneer Park. All of the examined elements were within the ranges proposed for European soils. The only exception was Pb in Hellbrunn Park in Salzburg, the levels of which were close to those reported for polluted urban soils [9,12,52,53]. This may be explained by the fact that, in the Salzburg area, mining sites that have been closed for hundreds of years still show the effects of high concentrations of certain potentially toxic elements [54].

### 3.3. Partitioning of Potentially Toxic Elements in Urban Soils

The optimised BCR four-step extraction procedure provides information on the main binding sites and the distribution of potentially toxic elements in different geochemical soil phases, with the successive use of extraction agents of varying aggressiveness. Elements released in the first three phases are considered most likely to be of anthropogenic origin, and potentially available for uptake by plants, while those present in the fourth, residual fraction are not available to either plants or microorganisms [24,32,33,45,47].

Since Al represents one of the most abundant elements of the Earth’s crust, its total soil content is most often derived from the parent rock from which the soil was formed, as confirmed by the results of this study. Specifically, the highest concentration of Al was extracted in the residual fraction in stable crystalline form (78.91–90.76%), indicating that Al in this form and under alkaline conditions is very stable and unavailable for plants, and therefore does not present a risk in terms of phytotoxicity [43]. Less than 1% of Al was extracted in the acid-soluble/exchangeable fraction. In the third phase, which represents elements bound to organic matter and sulphides, and the second fraction, representing Fe/Mn crystalline oxides, Al accounts for less than 10%, with the exception of Hellbrunn and Kurgarten, where Al in the second fraction is a bound form (11.99–12.36%), as shown in Figure 2.

All As content in the soil at all of the sampling sites was found in the organic matter substrate (F3), which is to be expected given its strong sorption for clay particles and organic matter [43]. The lowest As content was measured in soils with a high sand content (Thessaloniki) and low organic matter content (Thessaloniki and Košutnjak (Belgrade)), and due to the prevailing alkaline conditions in the soil in these parks, the solubility of As and its affinity to build oxides with metals decreases, which is why it does not pose a potential risk to the examined urban parks (Table 1).

Chromium occurs naturally in various oxidation states, with trivalent chromite Cr(III) being the most stable form, and hexavalent chromate Cr(VI) being highly mobile in soil and extremely toxic to living organisms [45,55]. Since Cr(III) is poorly soluble in water and completely precipitated at pH values above 5.5 [56,57], it can be assumed that in the investigated soils, which are characterised by neutral-to-alkaline pH, Cr(III) is present in a very stable form [47]. This was further confirmed by sequential analysis, where the proportion of Cr at all of the investigated sites was highest in the residual fraction (71.20–83.92%). The rest was associated with organic matter and sulphides (10–20%), while only a small part was bound to Fe/Mn oxides (< 5%). The Cr content in the acid-soluble/exchangeable fraction was very low, i.e., below the detection limit (Figure 2). These results are indicative that Cr has a strong association with the insoluble fraction and, thus, that it is chemically stable and biologically unavailable for uptake by plants.

Copper in soil has a great ability to interact with other minerals, the clay fraction, organic matter, and iron and manganese oxides, but it is also dependent on its content in the parent material [5,43]. Sequential extraction showed that in the soils in the Košutnjak, Lagadas, and Hall Pioneer parks, the highest percentage of Cu (60%) was bound to the crystalline structure of the minerals, while the rest was associated with organic matter and sulphides, i.e., with Fe and Mn oxides. Only a small proportion of Cu (<5%) was associated with the acid-soluble/exchangeable fraction (Figure 2). A somewhat different distribution of Cu was found in soil samples in the Hellbrunn and Kurgarten parks, where the percentage of Cu extracted in the oxidisable fraction was about 50%, which makes this element potentially mobile and bioavailable. The tendency of Cu to bind to organic matter and sulphides has also been noted by other authors [9,58,59], and can be explained by the fact that the complex thus formed has great stability due to the high affinity of Cu(II) to soil organic matter [47,58]. The presence of Cu in the oxidisable and reducible fractions may also reflect anthropogenic pollution in surface soils, since it represents one of the most serious environmental contaminants, released from vehicles and from processes in the metal industry [60,61].

Concentrations of Ni at the unpolluted sites of Hellbrunn and Košutnjak, and the polluted sites of Aristotelous Square and Hall Pioneer Park, were higher than the average content of this element for worldwide soils (13–26 mg kg^−1^, [43]), but within the background values for European soils (30–75 mg kg^−1^, [51]). However, sequential analysis showed that the majority of Ni, despite its elevated content in the examined soils, was extracted in the residual fraction (58.44–67.51%), indicating its low mobility and availability. The smallest portion was associated with the acid-soluble/exchangeable fraction (5–7% in Aristotelous Square, Košutnjak, Lagadas, and Hall Pioneer Park), while in soil samples from Hellbrunn and Kurgarten it was not extracted at all in the first fraction (Figure 2). The remaining Ni was distributed equally between Fe/Mn oxides and the organic matter substrate, due to the high affinity of Ni to organic matter, which is why this element is highly concentrated in coal and oil [62]. The results of BCR extraction showed that, irrespective of the concentration of Ni in soils—which is usually the result of the natural enrichment of soil during pedological processes—anthropogenic sources of Ni from industry, thermo-energy facilities, traffic, etc., should not be neglected [56].

Sequential extraction of Pb revealed a significantly different phase distribution compared to the other elements, primarily due to the high percentage of Pb in the reducible fraction—i.e., bound to Fe and Mn oxides (37–51%)—and also to the residual fraction (17–51%), as shown in Figure 2. These results coincide with earlier findings that most Pb is associated with the reducible fraction, with very low amounts present in the exchangeable fraction in polluted soils [63]. The tendency of Pb to be associated with Fe and Mn oxides has also been widely reported in other studies in urban regions [12,47]. However, a higher percentage of Pb content in the acid-soluble/exchangeable fraction (up to 28%) was found in the soils in the Hellbrunn, Kurgarten, and Hall Pioneer parks. This fractional profile of Pb indicates that its presence in the examined soils is primarily the result of emissions from motor vehicles and the presence of suspended particles originating from industry [8,9]. The fact that Pb most likely originates from traffic is supported by the statistically significant correlation with Cu and Zn—elements released into the urban atmosphere mostly by vehicles and fuel combustion [60]. For pH values >8, Pb precipitates into the soil in the form of lead hydroxide, phosphate, or carbonate, forming stable organic complexes [43]; however, if weakly acidic conditions are established, as well as changes in redox potential and salinity, Pb can be released from the reducible fraction and become available, indicating an environmental risk [57].

The Zn fractional profile showed that it originates from both natural and anthropogenic sources. The highest percentage of Zn was in the residual phase in all of the studied soils (42.14–54.65%), indicating its lithogenic origin (Figure 2). However, the share of the first three fractions in the profile is not negligible, which may be related to the anthropogenic influence, particularly to motor vehicle emissions and other waste products of traffic. Similar Zn behaviour in urban soils has been reported in other studies [59,64]. A significant proportion of Zn associated with carbonates and Mn and Fe oxides (66.62%) was measured in soil samples around major urban roads in Belgrade with low organic matter contents, which is why Zn is predominantly bound to the inorganic and soluble organic soil fractions [57]. Based on the results of BCR analysis, it can be concluded that Zn in the examined urban soils is of both natural and anthropogenic origin, and that any change in environmental conditions (e.g., pH, redox potential, etc.) can cause its remobilisation and increase its availability for plant uptake.

Sequential analysis showed that Al, Cr, and Ni had strong structural connections with the crystalline lattice of soil minerals, indicating their predominantly lithogenic origin and their low mobility in the studied soils, especially in Aristotelous Square and Hall Pioneer Park. In contrast, Pb, Zn, and to a lesser extent Cu were affected by anthropogenic factors in addition to their lithogenic origin. Their percentages in the first three fractions were equal to or even higher than in the residual phase in some localities. In the Kurgarten and Hellbrunn parks, Pb content was above the European and world averages, and there was also found to be a high percentage of Pb in the acid-soluble/exchangeable fraction (24–28%), which can pose a significant environmental problem. Similarly, more than 10% of Zn in the acid-soluble/exchangeable fraction was found in Kurgarten (12.23%) and Hall Pioneer Park (13.22%), where the total Zn content was also above the global range for this element.

A Q-mode cluster analysis (CA), which processes sites, was performed in order to find similarities between the soils at all of the examined sites, taking into account the phases of sequential extraction. Figure 3 shows the dendrograms for the investigated sites for each phase of sequential extraction. Two different and strong associations can be identified for the first phase (the acid-soluble/exchangeable phase, and of most interest from the environmental point of view). The first connects Aristotelous Square, Košutnjak, Lagadas, and Hall Pioneer Park, indicating similarities in the metal distribution of the exchangeable phase, while the second association indicates similarities between the soils from Hellbrunn and Kurgarten. There are similarities between two groups of soils in terms of the Fe/Mn crystalline substrate: (1) Košutnjak, Hall Pioneer Park, and Aristotelous Square, and (2) Hellbrunn, Kurgarten, and Lagadas; and three groups when it comes to organic matter and sulphides: (1) Košutnjak and Hall Pioneer Park, (2) Hellbrunn and Kurgarten, and (3) Aristotelous Square and Lagadas, indicating differences in the vegetation of their regions. Finally, two types of strong crystalline structures are indicated in the residual phase: similarities can be found for (1) Aristotelous Square, Kurgarten, and Hellbrunn, and for (2) Košutnjak, Hall Pioneer Park, and Lagadas.

### 3.4. Identification of Sources of Potentially Toxic Elements in Urban Soils

In this study, PCA of PTEs in urban soils showed two principal components (PC1 and PC2) that explained 80.10% of all data variation. Previous studies have shown the impact of traffic pollution on the accumulation of PTEs in urban soils, primarily resulting from vehicle emissions. In particular, Pb, Cu, and Zn are mainly associated with vehicles, with Pb resulting from the combustion of leaded fuel, Cu from brakes, and Zn from the wear and tear of tyres, while sources of Cr and Ni include heating plants, the metallurgical industry, and traffic [9,60,65,66,67,68]. Thus, this was singled out as a particular factor in our study. However, Ni and Cr can also occur naturally in the soils considered for this study, as they may be derived from the geological substrate. Therefore, in the above analysis, factor PC1 is defined as the anthropogenic factor, mostly related to urban traffic and industry, while PC2 relates to the natural, lithogenic factor. PC1 explains 60.71% of variability, and singles out As, Cu, Pb, and Zn as elements characteristic of urban zones with heavy traffic. PC2, explaining 19.39% of variability, singles out Cr and Ni, together with Al, which is used as a background element, confirming the lithogenic origin of these elements.

Spatial distribution based on PC1 distinguishes the sampling sites of the Hellbrunn and Kurgarten parks in Salzburg (Figure 4) as being predominantly influenced by anthropogenic factors. The lithogenic origin of Cr and Ni is evident in Aristotelous Square Park in Thessaloniki, as well as in Hall Pioneer Park in Belgrade, which is confirmed by the high share of these elements in the residual fraction, in particular of Cr (Figure 2); however, their positioning towards PC1 also indicates an anthropogenic impact on the accumulation of these elements in soil. Namely, in central parts of Serbia, including Belgrade, the origin of Ni and Cr in soil is determined by the geological substrate. Previous studies conducted by Jakovljević et al. [69] and Čakmak et al. [20] showed an increased content of Ni and Cr in soil formed on serpentine rocks in western Serbia and in the valleys of large rivers, where they originate from pedogenetic processes of alluvium formation. Similarly, Saljnikov et al. [70] found an average concentration of Ni of 75.75 mg kg^−1^, and of Cr of 57.8 mg kg^−1^, in the soils of large river valleys in Serbia. Serpentine rocks are also a significant feature of the geology of the Thessaloniki area [71], because a serpentine (ophiolitic) substrate covers large areas of the region. Serpentine substrates are characterised by high concentrations of Ni, Cr, Co, Pb, Zn, Al, Fe, Mg, and Cd [72,73]. In the ultramafic soils of the Thessaloniki area, Ni content ranges from 110 to 2700 mg kg^−1^ [71]. In terms of Ni and Cr, as well as the content of other elements in soil samples, the sites of Lagadas (Thessaloniki) and Košutnjak (Belgrade) stand out from the other PC1/PC2 components, i.e., their origin is not related solely to anthropogenic factors or to the lithogenic factor, but to both (Figure 4). The high levels of elements—particularly Pb, Zn, and As—measured in soil samples from Salzburg, both in the city centre and in the suburban area, might be associated with anthropogenic emissions [48].

Cr, Ni, Pb, Zn, Ba, and Cd are elements of anthropogenic origin, originating from traffic, industry, and other anthropogenic sources [74,75]. The comparatively higher concentrations of Pb and Zn in the cities are likely to be the result of the large number of vehicles [74]. The type and intensity of traffic generates trends in atmospheric pollution in urban environments that are more evident in urban centres than in residential areas. In this study, at the polluted sites, the number of vehicles per day was in the following order: Thessaloniki, with 57,061 vehicles passing through Aristotelous Square > Salzburg, with 38,000 passing by Kurgarten park > Belgrade, with 23,237 vehicles passing by Hall Pioneer Park per day (data were provided by city public transportation departments). Fuel, especially diesel, contains Fe, Mg, Cr, Ni, Pb, Zn, Ba, and Cu, and additives to lubricant oils also influence the chemical composition of exhaust gases [68]. Aluminium is linked to the wear and tear of construction materials, wood burning, and coal combustion, while Ni and Cr are metals related to industrial activities, waste disposal, etc. [27,76,77]. However, the results obtained in this study show that Cr and Ni are likely to be of natural origin due to the specific geological substrate, which is abundant in these elements, while Pb, Cu, and to some extent As are related to anthropogenic factors.

The highest Cu and Ni contents were measured in Aristotelous Square, while Kurgarten exhibited higher concentrations of As, Pb, and Zn than those measured in the Hall Pioneer and Aristotelous parks. The high levels of PTEs—particularly As, Pb, and Zn—measured in soil samples from Salzburg, both in the city centre and in the suburban area, might be associated with particulate emissions produced by fuel combustion, as well as the impact of mining. An earlier soil survey showed that As levels were especially high in some regions of Salzburg, Styria, and Lower Austria due to lithogenic or anthropogenic sources, but that Pb content also exceeded the target value of 100 mg kg^−1^ in Salzburg [48], which is further confirmed by our research. In absolute terms, the mean Pb concentration in this city, although higher than the background levels proposed for European soils, is comparatively low in comparison to other cities; for instance, it is lower than the mean Pb concentration found in Paris (>200 mg kg^−1^, [53]), Glasgow, Seville, Torino (183–221 mg kg^−1^, [12]), Palermo (137–443 mg kg^−1^, [52]), Naples (262 mg kg^−1^, [9]), and Moscow (174 mg kg^−1^, [78]).

### 3.5. Health Risk Assessment

The calculated non-carcinogenic (HQ, HI) and carcinogenic (CR, TRC) indices for both children and adults are presented in Figure 5. The non-carcinogenic risk values for children decreased in the following order: Pb > As > Ni > Zn > Cu > Cr; and for adults: Pb > As > Ni > Cu > Zn > Cr. Lead was the dominant contributor to the non-carcinogenic risk for both age groups.

The HI values obtained for both children and adults were less than 1, indicating that there is not a non-carcinogenic risk from PTEs. However, HI values were higher for children than for adults, which implies that children are more sensitive to the impact of PTEs due to their habits of oral intake. The highest HI values were calculated for Hellbrunn and Kurgarten, where Pb and As of anthropogenic origin (as proven by PCA analysis) had the greatest influence.

The carcinogenic risk (CR) was calculated for As, Pb, and Ni, given the fact that reference doses or conversion factors for Cr(III), Cu, and Zn are unavailable [36]. Similar to the non-carcinogenic risk, CR values for children and adults were obtained based on three types of exposure (dermal, ingestion, and inhalation). The carcinogenic risk values decreased in the following order: As > Pb > Ni; however, CR results obtained for As were higher than for the other elements, which is in line with previous studies [35,37,79]. Although the CR for As was higher than for the other tested elements, it does not represent a health risk, since the calculated CR values are classified as an acceptable total risk. Overall, the results obtained in this study indicate that the TCR for both children and adults is considered acceptable at all sites, but that the Hellbrunn and Kurgarten sites pose a higher risk, probably due to elevated concentrations of As and Pb in the soil at these localities.

## 4. Conclusions

Topsoil samples from polluted urban parks in the cities of Salzburg, Thessaloniki, and Belgrade revealed levels of Al, Cr, Cu, Ni, Pb, and Zn that, for the most part, were higher than those in unpolluted soils, except at Hellbrunn Park. Based on the whole dataset, it can be concluded that As, Cu, Pb, and Zn derive from anthropogenic sources, whereas Al, Cr, and Ni distributions are mainly controlled by lithogenic inputs.

Results of sequential analysis showed Al, Cr, and Ni had strong structural connections with the crystalline lattice of soil minerals, proving their predominantly lithogenic origin and their low mobility in the studied soils, especially in Aristotelous Square and Hall Pioneer Park. In contrast, the influence of the anthropogenic factor on Cu, Pb, and Zn is evident, especially at the Kurgarten and Hellbrunn parks.

Similarly, PCA analysis confirmed the BCR results for Aristotelous Square and Hall Pioneer Park, proving the dominant influence of natural factors and the lithogenic origin of Ni and Cr in soils, as well as both lithogenic and anthropogenic inputs of other elements. In Salzburg, the anthropogenic origin of As and Pb is evident at both the unpolluted (Hellbrunn) and polluted (Kurgarten) sites, which points to anthropogenic activities, especially at Hellbrunn, probably due to the impact of historical pollution from mining.

Site-dependent variations showed that the highest concentrations of As, Cu, Pb, and Zn of anthropogenic origin were recorded in Salzburg, while the highest levels of Al, Cr, and Ni of lithogenic origin were recorded in Belgrade and Thessaloniki, which reflects the specificity of the geological substrate. Bearing in mind, however, that the content of the tested PTEs was below the values for European soils, it is evident that PTEs do not represent a potential risk to the examined urban soils. Nevertheless, any change in the environment (redox potential or soil pH and salinity) could lead to significant changes in the fractional profiles of the studied elements, which could pose a serious environmental problem, especially in the case of Pb and Zn. The results of the combined analyses and the distribution patterns of the pollutant metals suggest that motor traffic represents the most significant anthropogenic pollutant source for the studied urban environments.

Results obtained for the health risk assessment showed that no non-carcinogenic or carcinogenic risk was found for either children or adults for any of the investigated elements. However, future research requires methodological innovation and international standardisation so that the multiple issues related to environmental health can be tackled and the research methods associated with environmental health data and risk assessment methods can be improved.

## Figures and Tables

**Figure 1 ijerph-18-06014-f001:**
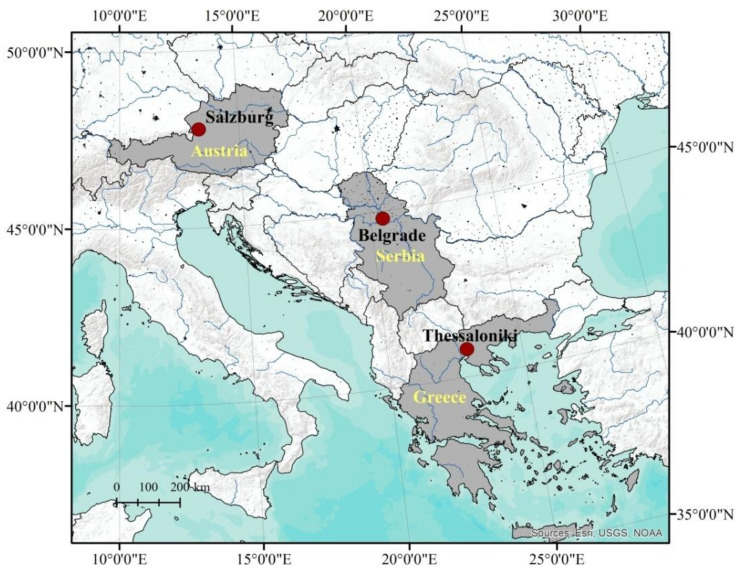
The study sites in Salzburg, Belgrade, and Thessaloniki.

**Figure 2 ijerph-18-06014-f002:**
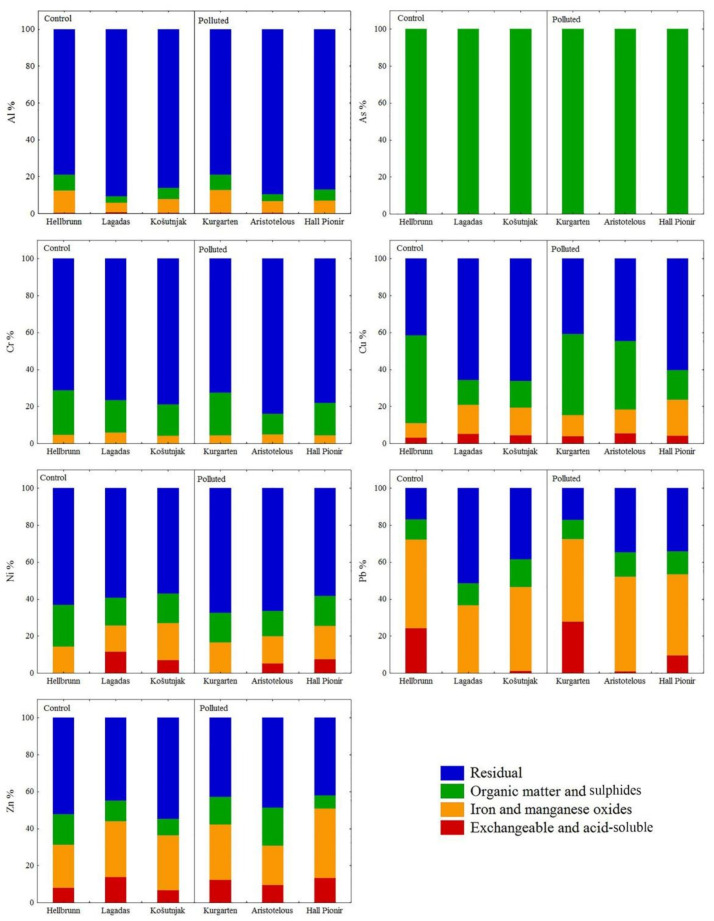
Partitioning of Al, As, Cr, Cu, Ni, Pb, and Zn in urban soil samples. Proportions of elements in the acid-soluble/exchangeable phase, bound to Fe/Mn oxides, bound to organic matter, and in the residual phase.

**Figure 3 ijerph-18-06014-f003:**
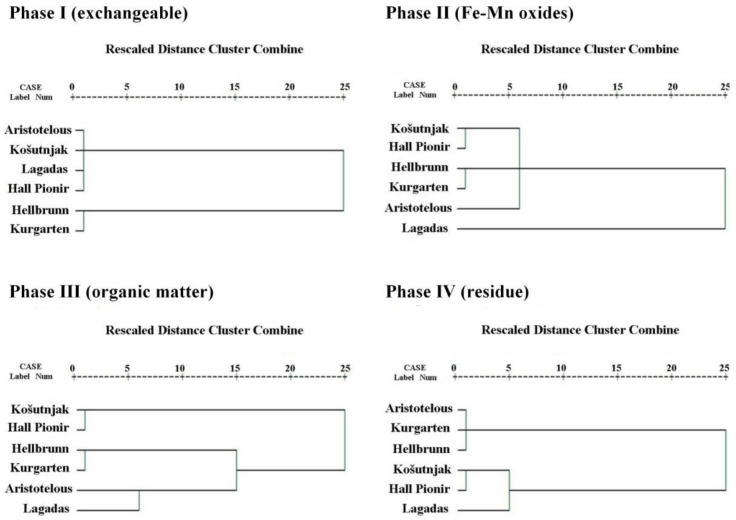
Q-mode cluster analysis dendrograms of the cumulative sub-datasets representing the sequential steps for all of the sites.

**Figure 4 ijerph-18-06014-f004:**
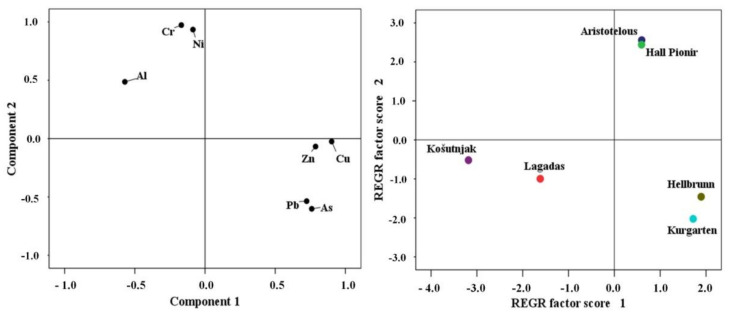
PCA loading plot for urban soils. Graphic score for soil at the examined sites.

**Figure 5 ijerph-18-06014-f005:**
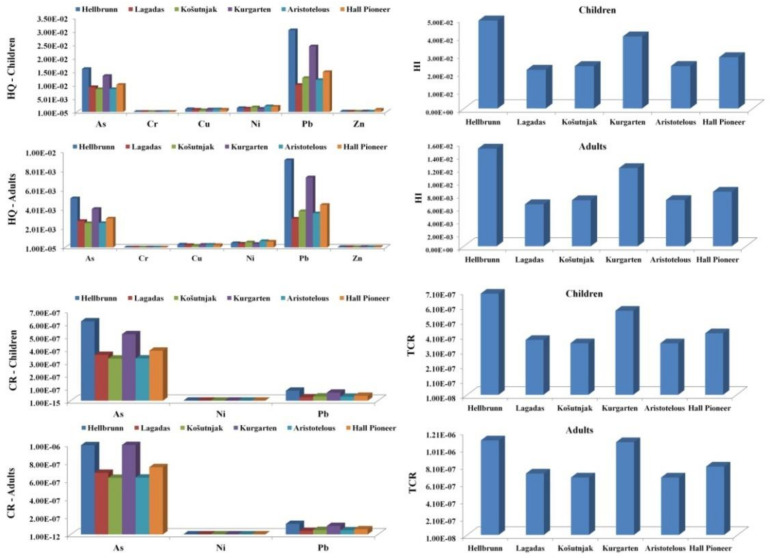
Non-carcinogenic (HQ, HI) and carcinogenic (CR, TRC) indices for children and adults from PTEs in soil.

**Table 1 ijerph-18-06014-t001:** Selected physical and chemical parameters of sampled urban soils.

Sample	pH (H_2_O)	Sand	Clay	Silt	Soil Organic Matter	* OC	Soil Type
		%					
Control sites							
Hellbrunn	6.78	43.81	7.93	48.26	7.16	4.16	Silt loam
Lagadas	8.02	63.07	11.34	25.59	1.58	0.92	Sandy loam
Košutnjak	7.90	20.03	30.04	49.93	2.07	1.20	Clay loam
Polluted sites							
Kurgarten	6.91	58.60	6.62	34.78	7.03	4.08	Sandy loam
Aristotelous Square	8.06	62.18	9.90	27.92	1.52	0.88	Sandy loam
Hall Pioneer	8.47	53.28	26.40	20.32	3.94	2.28	Sandy clay loam

* OC—Organic carbon.

## Data Availability

Not applicable.

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
