# Peer review of "Fractionation of Potentially Toxic Elements (PTEs) in Urban Soils from Salzburg, Thessaloniki and Belgrade: An Insight into Source Identification and Human Health Risk Assessment"

_ijerph, 2021, doi:10.3390/ijerph18116014_

Round 1

Reviewer 1 Report

Overall a well-written paper.

The important conclusion 'no human health risk for both adults and children' is missing in the Abstract. This should be added.

The conclusions based on the results as presented in Table 2 need more attention. In particular the comparison of the measured data from this study with the mean values world-wide and the European background data. More explanation should be given on these world wide and European reference data. For example, what is meant by 'background'? Is that a natural  background (i.e. non-antropogenic) or a more 'ambient' one (i.e. natural plus antropogenic, but not hot spot polluted) ? Measured data from this study are nearly all (far) below these European background levels. From that angle one may conclude that all soils in this study (whether called polluted or unpolluted here) are rather clean! To be honest, the distinction that was made beforehand between polluted and unpolluted soils in this study is rather weak: the differences in PTE levels are small despite statistical testing (SDs are also very, surprisingly small !!). Furthermore and importantly comparing metal levels with reference (background) data without any soil speciation correction (e.g. % organic matter) is speculative. In addition, the reason that global ranges are shown to be much lower than European background values should be explained as well. 

Minor: why is the element cadmium missing?

Reviewer 2 Report

This piece of research is very timely and I would recommend it to be accepted to be published after the following minor edits.

Introduction

Lines 46-51 seemed to be a repetition of lines 35 to 40. I strongly suggest eliminating repetitions and rewriting the introduction.

Materials and Methods

Line 104 - This sentence states " winters are much cooler". Please specify what are the winter months prior to this statement as winter months are different from country to country.

Lines 163-166 - It is advisable to include the continuing calibration verification (CCV) value samples at every 10th sample for quality assurance and quality control purposes in the future.

Results and discussion

Table 1 - Please consider significant figures when reporting the data. Reporting soil organic matter and OC values to two decimal points or less will be more appropriate. Please include footnotes under the table to specify what is meant by the abbreviations used in the table.

Lines 301-303 - Better to elaborate a bit about he BCR test which was used to obtain the results stated under the section 3.3, at the beginning of that section.

Lines 471-474 - Please state the reference to this sentence.

Lines 475-476 - Please don't break a scientific name across two sentences. E.g. Write Aluminium instead of Al-uminium

Line 514 - Please don't start a sentence with an abbreviation.

For example write The carcinogenic risk instead of CR, when the sentence begins with that.

Reviewer 3 Report

The research is interesting, the authors establish mechanisms to carry out sequence (time) of the risk due to contamination of heavy metals in urban areas, as well as the factors involved.
